# Biopsychosocial Response to the COVID-19 Lockdown in People with Major Depressive Disorder and Multiple Sclerosis

**DOI:** 10.3390/jcm11237163

**Published:** 2022-12-01

**Authors:** Sara Siddi, Iago Giné-Vázquez, Raquel Bailon, Faith Matcham, Femke Lamers, Spyridon Kontaxis, Estela Laporta, Esther Garcia, Belen Arranz, Gloria Dalla Costa, Ana Isabel Guerrero, Ana Zabalza, Mathias Due Buron, Giancarlo Comi, Letizia Leocani, Peter Annas, Matthew Hotopf, Brenda W. J. H. Penninx, Melinda Magyari, Per S. Sørensen, Xavier Montalban, Grace Lavelle, Alina Ivan, Carolin Oetzmann, Katie M. White, Sonia Difrancesco, Patrick Locatelli, David C. Mohr, Jordi Aguiló, Vaibhav Narayan, Amos Folarin, Richard J. B. Dobson, Judith Dineley, Daniel Leightley, Nicholas Cummins, Srinivasan Vairavan, Yathart Ranjan, Zulqarnain Rashid, Aki Rintala, Giovanni De Girolamo, Antonio Preti, Sara Simblett, Til Wykes, Inez Myin-Germeys, Josep Maria Haro

**Affiliations:** 1Parc Sanitari Sant Joan de Déu, Fundació Sant Joan de Déu, CIBERSAM (Madrid 28029), Universitat de Barcelona, 08007 Barcelona, Spain; 2Aragón Institute of Engineering Research (I3A), University of Zaragoza, 50001 Zaragoza, Spain; 3Centros de Investigación Biomédica en Red en el Área de Bioingeniería, Biomateriales y Nanomedicina (CIBER-BBN), 28029 Madrid, Spain; 4Institute of Psychiatry, King’s College London, Psychology and Neuroscience, London SE5 8AF, UK; 5School of Psychology, University of Sussex, Falmer BN1 9QH, UK; 6Department of Psychiatry, Amsterdam UMC, Vrije Universiteit, 1081 HV Amsterdam, The Netherlands; 7Mental Health Program, Amsterdam Public Health Research Institute, 1081 BT Amsterdam, The Netherlands; 8Microelectrónica y Sistemas Electrónicos, Universidad Autónoma de Barcelona, 08193 Bellaterra, Spain; 9Faculty of Medicine, Vita-Salute San Raffaele University, 20132 Milan, Italy; 10Multiple Sclerosis Centre of Catalonia (Cemcat), Department of Neurology/Neuroimmunology, Vall d’Hebron Institut de Recerca, Hospital Universitari Vall d’Hebron, Universitat Autònoma de Barcelona, 08035 Barcelona, Spain; 11Danish Multiple Sclerosis Center, Department of Neurology, Copenhagen University Hospital Rigshospitalet, 2100 Copenhagen, Denmark; 12Casa Cura Policlinico, 20144 Milan, Italy; 13Experimental Neurophysiology Unit, Institute of Experimental Neurology-INSPE, Scientific Institute San Raffaele, 20132 Milan, Italy; 14H. Lundbeck A/S, 2500 Valby, Denmark; 15Department of Engineering and Applied Science, University of Bergamo, 24129 Bergamo, Italy; 16Center for Behavioral Intervention Technologies, Department of Preventative Medicine, Northwestern University, Chicago, IL 60611, USA; 17Research and Development Information Technology, Janssen Research & Development, LLC, Titusville, NJ 08560, USA; 18Department for Neurosciences, Center for Contextual Psychiatry, Katholieke Universiteit Leuven, 7001 Leuven, Belgium; 19Faculty of Social Services and Health Care, LAB University of Applied Sciences, 15210 Lahti, Finland; 20IRCCS Instituto Centro San Giovanni di Dio Fatebenefratelli, 25125 Brescia, Italy; 21Dipartimento di Neuroscienze, Università degli Studi di Torino, 10126 Torino, Italy

**Keywords:** COVID-19, SARS-CoV-2, major depressive disorder, multiple sclerosis, depression severity, heart rate, stress, social activity, physical activity, decentralized

## Abstract

Background: Changes in lifestyle, finances and work status during COVID-19 lockdowns may have led to biopsychosocial changes in people with pre-existing vulnerabilities such as Major Depressive Disorders (MDDs) and Multiple Sclerosis (MS). Methods: Data were collected as a part of the RADAR-CNS (Remote Assessment of Disease and Relapse—Central Nervous System) program. We analyzed the following data from long-term participants in a decentralized multinational study: symptoms of depression, heart rate (HR) during the day and night; social activity; sedentary state, steps and physical activity of varying intensity. Linear mixed-effects regression analyses with repeated measures were fitted to assess the changes among three time periods (pre, during and post-lockdown) across the groups, adjusting for depression severity before the pandemic and gender. Results: Participants with MDDs (N = 255) and MS (N = 214) were included in the analyses. Overall, depressive symptoms remained stable across the three periods in both groups. A lower mean HR and HR variation were observed between pre and during lockdown during the day for MDDs and during the night for MS. HR variation during rest periods also decreased between pre- and post-lockdown in both clinical conditions. We observed a reduction in physical activity for MDDs and MS upon the introduction of lockdowns. The group with MDDs exhibited a net increase in social interaction via social network apps over the three periods. Conclusions: Behavioral responses to the lockdown measured by social activity, physical activity and HR may reflect changes in stress in people with MDDs and MS. Remote technology monitoring might promptly activate an early warning of physical and social alterations in these stressful situations. Future studies must explore how stress does or does not impact depression severity.

## 1. Introduction

In January 2020, the World Health Organization (WHO) [1] declared the new coronavirus SARS-CoV-2 epidemic, which caused COVID-19 to be a Public Health Emergency of International Concern. The first cases were identified in Wuhan, Hubei Province, China, in late December 2019. On 11 March 2020, the WHO declared the COVID-19 outbreak a pandemic [1]. When the virus appeared, several countries imposed a national lockdown to limit the spread of the virus, with restrictions varying from country to country. The government in Italy announced a nationwide lockdown on 9 March 2020 [2].

The lockdown was implemented in Denmark and Spain one week later on 13 March [3] and 14 [4], respectively, but with differing restrictions. Other countries, such as the UK [5] and the Netherlands, implemented a lockdown in late March [6]. The lockdown measures during the COVID-19 outbreak were expected to have adverse consequences, including financial losses, scarcity of basic supplies, family separations, and an increased perception of risk [7,8,9,10]. This pandemic has increased the necessity to strengthen mental health systems; the cases of Major Depressive Disorders (MDDs) increased globally by 27.6% [11]. Google Trends analysis revealed a massive rise in searches related to symptoms of depression and stress during the COVID-19 pandemic [12]. These negative consequences could represent an even higher toll on people who already have a pre-existing mental illness [13]. The psychological effects of the COVID-19 pandemic reported were a reduction in face-to-face social contact [14], increased levels of loneliness [15,16,17], psychological distress [18], changes in depression levels [7], decreased self-esteem and sleep duration [14,19] and, in extreme cases, suicide [8].

The literature has widely documented the impact **of** stressful life events on major depressive disorders (MDDs) [20,21] and also on Multiple Sclerosis (MS) [22,23]. MS disease has high comorbidity with depression [24]. Consistent evidence shows high prevalence rates of depression (31%) and anxiety (22%) in MS [24]. The restrictions required by the lockdowns may also have worsened depression symptoms in patients with MS. A review reported that individuals with MS are likely to feel depressed due to their psychosocial circumstances [25]. They felt helpless, had low-quality social relationships and high levels of stress, and adopted maladaptive coping strategies. Psychosocial factors related to a lack of social support, adverse experiences, and life events may significantly impact women more than men [26]. Although there is a large body of literature on the psychological effects of COVID-19, most of it has selected convenient samples and has used non-standardized instruments for the diagnosis [18]. There is also limited information on the impact on individuals that suffer from depression or other chronic conditions; COVID-19 might have exacerbated pre-existing clinical symptoms in people with MDDs and MS. Another study found an increase in depression severity in patients with MDDs caused by the pandemic, supported by a higher number of hospitalizations of patients with MDDs at a short-term acute care hospital, and all-cause inpatient hospitalizations at a psychiatric facility increased following the pandemic onset and remained elevated throughout the first six months [27].

Remote measurement technology (RMT) allows data collection in real time and a natural environment without high costs. It does not require face-to-face contact between the research team and participants. Decentralized (“virtual”) research with RMTs provides a large quantity of data over a period without intruding into the daily life of research participants [28]. The RADAR-CNS (Remote Assessment of Disease and Relapse—Central Nervous System) consortium developed the open-source mHealth platform RADAR-Base [29] to collect longitudinal data using RMT (phone and activity trackers), providing high-frequency data on depressed mood, self-esteem, speech, and cognition, and passively on heart rate, physical activity, sleep, and sociability. Previous analyses of this large sample were conducted to explore the impact of the pandemic on behavior (physical, sleep, social activity), focusing on the differences by country [14], and another focusing on the exploration of depression, self-esteem, and sleep in people with MDDs [19]. Both studies demonstrated the validity and utility of the RADAR-Base to detect changes during the pandemic in the studied sample. 

Considering the lockdown restrictions, increased levels of depression, reduced physical activity, and intensified social interaction via phone were expected. In this decentralized study, we also explored the photoplethysmography (PPG)-based heart rate (HR) series provided by a commercial activity tracker.

The variability of HR is often used as an indicator of autonomic nervous system (ANS) regulation of the heart [30,31,32]. HR can vary significantly over 24 h in different conditions [33]. Increased HR variability generally reflects a healthy ANS function that can respond to changing circumstances [34], while low HR variability is a sign of a monotonously regular HR [35,36]. HR variability is altered in people with depression [37] as they could have difficulties in psychological arousal regulation in the presence of emotional or environmental stressors [38]. Dysfunctions in HR variability have also been found in people with MS [39,40]. Alterations in HR could be an indicator of stress [41]. Since MDDs and MS have also been linked to alterations in HR, these alterations may have influenced their stress response.

The main objectives of this study were to explore the impact of the lockdown, exploring the changes in depression level, physical activity, HR and sociality across three COVID-19 periods (pre-lockdown, during- lockdown and post-lockdown), and to understand if changes depend on depression severity before the pandemic in people with MDDs and MS. We also explored the impact of gender on the previous measures. 

## 2. Materials and Methods

Digital data were collected as part of the international research consortium RADAR-CNS (https://www.radar-cns.org/ (accessed on 18 January 2021)), a decentralized collaborative research initiative aiming to provide real-time multidimensional indicators of the clinical states of individuals with different health conditions: MDD, MS and epilepsy. In this manuscript, we focus our analyses on two health conditions: MDD [42] and MS [43]. Participants were recruited in five European countries: people with Major Depressive Disorder (The Netherlands, Spain and the UK) and people with Multiple Sclerosis (Italy, Denmark and Spain). Centers and countries were chosen based on their expertise and recruiting capacity for the study, with the aim of having a broad representation of different European geographies, both for the different disorders and the RADAR methodology.

The recruitment and follow-up started in November 2017 and finished in March 2021. For this study, we focused on the following periods (from December 2019 until June 2020, approximately)
During lockdown: we chose the entire period of the national lockdown in each country.For the pre-lockdown phase, we chose the period immediately before the first restrictive measure with the same duration as the total national lockdown.We chose the period immediately following the national lockdown for the post-lockdown phase with the same duration.

The following dates were considered to define the period:Denmark: lockdown began 13 March 2020, and post-lockdown started 16 April 2020Italy: lockdown began 9 March 2020, and post-lockdown started 19 May 2020Spain: lockdown began 14 March 2020, and post-lockdown started 5 May 2020The Netherlands: 15 March 2020 post-lockdown and started 12 May 2020UK: lockdown began 23 March 2020, and post-lockdown started 12 May 2020

The regional and local impact of the COVID-19 crisis has been highly heterogeneous, with significant implications for crisis management and policy responses [44], causing physical distance, loss of familiar and social connections in person and worsening mental health [45]. Different restrictions were implemented across countries with the requirement of staying at home except for essential trips for work, and all public events were cancelled, except in Denmark, where the measure was only recommended. Italy and Spain had more restrictive measures than other countries [46], with restrictions on gatherings and school closures. Places of work were required to close in some sectors in Spain, the United Kingdom, and Denmark, and were required to close for all but public services (stores, libraries, museums, etc.) in the Netherlands and Italy. Public transport was recommended to close in Italy, Spain, and Denmark [14]. However, the control measures of the pandemic may have caused a bidirectional effect: benefits in the control of the infection at the expense of a significant psychological impact. 

### 2.1. Procedure 

All local Ethics Committees approved the protocol of the institutions involved in the study, and participants provided written consent before enrollment (See [42] for MDD and [43] for MS).

### 2.2. Ethical Statement

The authors assert that all procedures contributing to this work comply with the ethical standards of the relevant national and institutional committees on human experimentation and with the Helsinki Declaration of 1975, as revised in 2008. All procedures involving patients for the study on MDD were approved by the Camberwell St Giles Research Ethics Committee (REC reference: 17/LO/1154), in London, from the CEIC Fundació Sant Joan de Deu (CI: PIC-128-17), in Barcelona, and from the Medische Ethische Toetsingscommissie VUms (METc VUmc registratienummer: 2018.012–NL63557.029.17), in the Netherlands; and for the study on MS, from Videnskabsetiske Komitéer (Region Hovedstaden. journal-nr: 18001543), in Denmark, from the CEIm of Vall d’Hebron Hospital (ID-RTF065) and the CE of the Ospedale San Raffaele (CE OSR: 255/2017). RADAR-CNS was conducted per the Declaration of Helsinki and Good Clinical Practice, adhering to principles outlined in the NHS Research Governance Framework for Health and Social Care (2nd edition). All participants provided informed consent to participate. All participants were under the standard treatments

### 2.3. Instruments

We analyzed the following features derived from the open-source mHealth platform RADAR-Base [29]. RADAR-base is developed as a modular application that includes active and passive apps for data collection. The active app includes questionnaires on mood, self-esteem, cognitive function and speech tasks. The passive app, which does not require active engagement, continuously collects data on a 24/7 basis through a smartphone and a Fitbit wrist-worn device, which includes phone usage, location, Bluetooth, heart rate and physical activity. For Fitbit devices (Fitbit Inc, San Francisco, CA, USA), Fitbit Charge 2/3 devices were given to participants. Participants were asked to wear this wearable device on their wrist of the non-dominant hand for the duration of the follow-up across cohorts, who provided ongoing information about physical activity and HR derived from the accelerometer and the PPG signals, respectively. This device was selected by the RADAR-CNS project members, involving the patients, who decided to use a commercially available device that is minimally intrusive and easier to use [47,48,49,50]. For this study, we focused on the analyses of the heart rate, physical activity captured using a wrist-worn device, and social activity and depression symptoms were captured by using a smartphone (Table 1).

Depressive symptoms: Depressive symptoms were assessed with the Patient Health Questionnaire (PHQ-8), which represents a valid instrument to identify depression severity in MDD and MS [51,52,53]. Participants with MDD were asked to complete the PHQ-8 [51] every two weeks, and participants with MS every six weeks through an app (active RMT)) [29] installed on an Android smartphone. The responses for each item vary from 0 = “not at all” to 3 = “nearly every day”. The PHQ-8 score ranges from 0 to 24 (increasing severity). A cut-off score of ≥10 is the most recommended cut-off point for “clinically significant” depressive symptoms, which means that the participant is likely to meet diagnostic criteria for a depressive episode (or moderate and severe depression) in the previous two weeks. Ratings below 10 are usually defined as asymptomatic or sub-threshold (no or mild depression). Internal consistency was calculated with Cronbach’s alpha, which was =0.91 for MDD and 0.88 for MS during the different assessments.

Social activity: Participants installed another app (passive RMT app) that collected data on behavioral activity via smartphone sensors, including social activity such as the number of contacts and app interactions. The data were collected continuously. Social contacts represent the number of new contacts added daily to the list of contacts concerning the previous measurement. The social interaction is represented by the sum of the number of interactions with social apps used to make calls, read and send messages per day through apps (i.e., Facebook, Messenger, Reddit, Skype, Telegram, Twitter, Viber, WhatsApp, etc.), measuring the activity of the user on these platforms.

Heart rate: In this study, we used the heart rate (HR) parameters reported to be related to depression severity in a previous study (Siddi et al.). Briefly, for assessing the HR profile, the mean level and standard deviation of the HR were computed for the whole day (24 h), during resting/sedentary periods during the day (24 h) and resting/sedentary periods at night (00:00 to 6.00). These data were collected with the Fitibit device, which was previously proven to measure HR accurately [32,54,55,56].

Physical activity and sedentary levels: Sedentary levels are estimated based on the mean number of steps per minute during the day, and physical activity is classified as sedentary, light, moderate and vigorous by proprietary algorithms of the Fitbit device. Physical activity was calculated by taking into account the time during which the individual was wearing the Fitbit. Periods without HR values are excluded from wear-time duration. Daily mean number of steps is a measure of global activity. Sedentary, light, moderate and vigorous activity were selected due to their strong association with depression [57,58,59] and during the pandemic (i.e., [60,61]).

### 2.4. Statistical Analysis 

First, we described the sociodemographic characteristics reporting frequencies and percentages for categorical variables, and the mean and standard deviation or medians with interquartile ranges (IQR) for continuous variables, as appropriate. We computed the average of each daily parameter (HR, physical and social activity) in the week before the PHQ-8 assessment across each pandemic-related period. 

Linear mixed-effects regression analyses for repeated measures (including depressive symptoms, HR, physical activity and social activity as independent variables) were fitted to assess the changes in the mean total score of each daily parameter among the three periods: pre-, during and post-lockdown (defined for each country). We then added the baseline depression severity into each model (Depression severe or moderate = 1, No depression or mild = 0, using a PHQ-8 cut-off ≥10). The baseline depression severity was calculated from the last PHQ-8 measurement before the pre-lockdown period, so it is generally in the range of the previous two weeks prior to that period. In this model, we also included an interaction term between period and baseline depression severity to investigate whether the change in the outcome variables over time varied according to depression severity (Figure 1 and Figure 2). Finally, in the previous model, we incorporated gender as a second exposure variable, including an interaction term between period, depression severity and gender, to investigate whether the rate of change in the outcome variables over time varied according to depression severity and gender.

All models incorporated a random effect for participants. All the analyses were conducted with R, packages Nonlinear Mixed Effects Models (nlme) [62] and Estimated Marginal Means (emmeans) [63].

## 3. Results

A total of 469 individuals, of which 71% were women, were included in the analyses: 255 people with MDD and 214 with MS (Table 2). We excluded participants with missing PHQ-8observations in the pre-lockdown interval (people with no data in the pre-lockdown as they were enrolled later or missed the assessment). Despite the large amount of digital data collected thanks to the RMT that monitors participants per day over a long period, missing values may occur due to technical problem*s* and daily life (levels of completeness have been reported in the Appendix A). For more details on the available data in the MDD cohort, see [64]. The mean age in the sample was 46.6-years-old (SD = 13.3). Half of the individuals in the MDD group had no partner (50.6%), while in the group with MS, most had a partner (65.6%). Years of education were higher in the MS group compared with the MDD group.

### 3.1. Summary of Findings in MDD

Table 3 displays the main findings across the three periods (pre, during and post-lockdown) in individuals with MDD. Overall, the mean depressive symptoms remained stable across these periods. 

The MDD group reduced their physical activity during the lockdown, as measured by the number of steps (*p* < 0.001), and it increased post-lockdown (*p* < 0.001) again. A similar pattern was found for light, moderate and vigorous activity, which reduced between the pre-lockdown and the lockdown (*p* < 0.001). Light and moderate activity also decreased between the pre-and post-lockdown periods (*p* < 0.05). All individuals reduced their sedentary levels from pre-lockdown to post-lockdown (*p* < 0.01). 

During the day, mean HR and HR variation reduced between pre and during lockdown (*p* < 0.001), while resting HR variation increased. The mean HR during the day (*p* < 0.001) and the resting state (*p* < 0.01) increased again between lockdown and post-lockdown. No differences were found for resting HR at night. 

We found a reduction in the HR variation (stdHR) all day (*p* < 0.05), especially during the resting state (*p* < 0.001) and an increase in mean HR in the resting period (resting mHR) (*p* < 0.001) during the day.

Overall, the social activity varied significantly during these periods. As we expected, the overall group intensified social interactions through apps, especially between pre and during the lockdown (Table 3).

#### Differences in Each Outcome between No or Mild Depression vs. Moderate or Severe Depression at Each Period and Interaction with Gender 

Figure 1 reports the differences between the groups with different depression severity (moderate or severe depression N = 124 versus no or mild depression at baseline N = 131). 

The group with moderate–severe depression at baseline reported higher depression levels in the three periods (*p* < 0.0001) (Appendix A) (Figure 1a). In the group with severe depression at baseline, women decreased the level of depression between pre and during lockdown compared to men (*p* < 0.05). (Appendix A).

No differences emerged for HR by depression severity pre and during the lockdown. The group with moderate or severe depression before the pandemic, compared to no or mild depression, reported lower HR variation all day (*p* < 0.05) and in the resting state (*p* < 0.05) during the post-lockdown. Regarding resting HR at night, in the group with higher depression levels before the pandemic, men decreased the resting mean HR during the night between pre- and post-lockdown compared to women (*p* < 0.05). 

The group with moderate or severe depression, compared to the mild or no depression group, reported a lower mean number of steps (Figure 1b) pre and during lockdown and vigorous activity during all periods (Figure 1f). No significant differences emerged for social activity (Figure 1m,n) (*p* > 0.05).

### 3.2. Summary of Findings for MS

Table 4 displays the main findings across the three periods (pre, during and post-lockdown) in the group with MS. In general, the mean depressive symptoms remained stable for the MDD group across these periods.

As measured by the number of steps, activity was reduced during the lockdown and increased from the lockdown to post-lockdown (*p* < 0.001). This was also observed for light activity between the pre-lockdown and during the lockdown (*p* < 0.001) and increased again between the lockdown and the post-lockdown (*p* < 0.05). Regarding HR, the most evident changes were observed between pre-lockdown and during the lockdown, and all HR parameters, either all day (*p* < 0.001) or at night (*p* < 0.05), were reduced. 

The resting mean HR increased after the lifting of lockdown, as evidenced when comparing during and post-lockdown periods (*p* < 0.01). We found a reduction in the HR variation (stdHR) all day (*p* < 0.05), especially during resting periods (*p* < 0.001), and reduced mean HR during the night (*p* < 0.01).

In the whole MS group, social activity remained stable, except for the social contacts, which increased from pre-lockdown to lockdown (*p* < 0.001) (Table 4).

#### Differences in Each Outcome between No or Mild Depression vs. Moderate or Severe Depression at Each Period and Interaction with Gender

Figure 2 (Please see Appendix A) reports the differences between the groups with different depression severity before the pandemic (moderate or severe depression N = 54 versus no or mild depression at baseline N = 160) across the three periods.

No significant changes emerged across all periods by depression severity, and no impact of gender on the studied variables was observed (Appendix A). The group with moderate or severe depression reported higher depression levels in all three periods (*p* < 0.001) (Figure 2a). 

The group with lower depression severity before the pandemic reported a higher mean number of steps during the pre-lockdown (*p* < 0.05) (Figure 2b) compared to the other group. No differences between the groups for light, moderate and vigorous activity (Figure 2c–f) and for social activity were found. No significant differences in gender emerged across these periods.

## 4. Discussion

Millions of people in different countries were confined in their homes due to the first wave of the COVID-19 pandemic, which led to unprecedented social, personal and psychological experiences. Various factors, such as the closure of schools, interruption of work in some cases, closure of recreational gyms, the restrictions on seeing relatives, and the recommendation or obligation to remain at home, drastically changed regular routines and may have potentially increased levels of depression. Long-term exposure to stress can negatively affect mental health and emotional well-being in individuals with long-term physical and mental health conditions [65]. Most publications in the literature explore the psychosocial effect of the lockdown on the general population [16,66]. The RADAR-CNS project offered the opportunity to explore the biopsychological and social impact of the lockdown and pandemic situations on the participants with MDD and MS who participated before and after the lockdown. The present work provides essential findings on psychosocial response measured by depression severity, social activity, physical activity and HR as a proxy measure of stress due to the lockdown in people with MDD and MS.

### 4.1. Findings on Participants with MDD 

Overall, the depressive symptoms remained stable across these periods. Previous studies have observed that lockdowns did not cause changes to depression levels in people who were already depressed before the pandemic [19,67,68]. However, depression in women with more severe depression decreased from pre-lockdown to lockdown compared to men, though this did not occur after the lockdown. The closure of education centers and nursery schools led to an increase in children’s care and household chores. This situation may have contributed to more significant differences between women and men during the lockdown in different communities [18,69,70,71,72,73]. 

Although women may experience higher stress levels due to their greater responsibility, the ability to use coping strategies could be related to a better adaptation to new stressful situations than men [74]. Another explanation might be that increasing social support during the pandemic may have enhanced women’s resilience [66]. Lockdown measures have increased social isolation and loneliness [15,16,17,75]. This could be one of the reasons why people with MDD showed intensified social activity measured through the social interactions with apps and the number of contacts across the three periods, especially during the lockdown. This aligns with our previous work [14], which demonstrated an increased proxy for social activity measured by Bluetooth connections during the lockdown. The group with MDD showed a reduced mean HR and HR variation during lockdown compared to the pre-lockdown period. A previous study showed that HR reduced during the lockdown [76], probably due to lower activity levels. We observed an increased resting mean HR between the lockdown and post-lockdown. From the pre- to post-lockdown, the patients with MDD had decreased the HR variation and increased the mean HR during rest. Significant HR variation reflects the capacity to adapt to new situations; depression was found to be associated with low HR variability in previous studies, especially during the resting period [37,77,78]. At the same time, we did not observe changes in the resting mean HR and HR variation during the night. In the group with more severe depression severity before the pandemic, men’s resting mean HR decreased during the night from pre- to post-lockdown compared to women. Men generally have a lower resting mean HR than women [79], though these differences may also vary with sleep quality, age and other factors such as the body mass index [80].

As expected, participants with MDD reduced their physical activity (all intensities) during the lockdown. Light and moderate activity also decreased from pre- to post-lockdown. The restrictions imposed limited movement outside, and all gyms were closed. These findings are confirmed by previous studies [14,76,81]. As we expected, the group with higher depression severity reported lower physical activity measured by the number of steps pre and during the lockdown, and vigorous activity during all periods with respect to the group with no or mild depression. Low physical activity levels are frequently observed in people with MDD [82,83].

### 4.2. Findings on Participants with MS

Depression is a common syndrome reported by people with MS [24,25], resulting in a low quality of life, increased fatigue and disability, and poor prognosis [25,84]. The stressful situation due to COVID-19 restrictions might have exacerbated depressive symptoms [23]. However, in our study, participants with MS did not report depression changes during the three periods. As we observed for MDD, the group with severe depression maintained the level of depression during all three periods. Other studies confirmed this absence of changes in people from different countries with MS [85]. 

We observed changes in the mean HR and HR variation all day and night, resting periods during the day and night, and activity. The decrement was observed between pre-lockdown and lockdown and between pre-lockdown and post-lockdown in daily mean HR and HR variation. A lower resting mean HR during the day may be related to the reduced physical activity during the lockdown; we observed a reduction in the number of steps in the same period as we observed for the group with MDD. 

The resting mean HR was reduced during the day and night from pre-lockdown and during the lockdown, but the resting mean HR during the day increased between lockdown and post-lockdown. An elevated resting mean HR between the lockdown and post-lockdown may be caused by other factors such as psychological stress perception [39], acute respiratory infections [55,86] or an increased alcohol intake ([87] et al. 2021). Overall, 21.8% of our participants with MS (total sample = 399) reported major symptoms similar to COVID-19 symptoms [43] and found an association with HR parameters [55]. Low HR variability was reported in people with MS [88], which may be caused by a significant increase in sympathetic cardiovascular tone [39] and can influence the course of the disease [40]. The reduction in the resting mean HR at night might be due to sleep alterations [39]. A previous study also reported sleep disruption in the same sample during the lockdown [14], which was confirmed in other populations [89]. Future studies are needed to explore whether HR parameters are altered due to other factors in this population. 

Consistent with our hypothesis, individuals with MS reduced their activity, as measured by the number of steps and minutes of light intensity of activity between pre and during lockdown, and increased between lockdown and post-lockdown. The group with lower depression reported a higher mean number of steps during the pre-lockdown. No differences were found in the different intensities of activity across the periods. It is well known that people with MS suffer from fatigue and gait dysfunctions [90]. This may be why we observe changes only in light activity between pre and during the lockdown, but not for moderate and vigorous activity, and between pre- and post-lockdown. The stressful situation caused by COVID-19 can add to the feeling of overwhelming fatigue [91]. In the whole MS group, the online social activity remained stable, except for the social contacts. They may also have perceived feelings of loneliness [92]. The lockdown increased social isolation for all people, but with different impacts on different groups.

### 4.3. Strengths and Limitations

The strengths of our study are that our sample was extensive in terms of observations and included participants from two different chronic diseases. Moreover, we had three periods: pre-lockdown, lockdown, and post-lockdown. Our main aim was to explore how the lockdown impacts two different health conditions by depression severity. Participants with MDD and MS answered **a** different number of depression questionnaire**s** (PHQ-8). Nonetheless, we included the participants that completed a minimum of two questionnaires. The data extracted by the PPG are highly correlated with the parameters of HR variability removed by the ECG [93,94]. However, wrist-worn devices seldom give access to the PPG signal, but instead to HR series derived by proprietary algorithms. Participants came from four European countries; differences were explored in our previous work [14]. We used a real-time assessment and conducted it in a natural environment. Our findings show that the RADAR-Base system could monitor psychosocial impact changes due to stressful life events. Various factors might have had an impact on the levels of depression: economic status, changes in contact with families, such as the restriction on seeing relatives face-to-face during the lockdown or the loss **of** physical demonstrations of affection, especially in countries that imposed stricter limitations than others, as was the case with Spain and Italy. Additionally, participating individuals may have received and changed treatment for depression during the course of the study, which may have influenced depression levels. Further decentralized research should consider these factors. Furthermore, as long as the technical details of Fitbit devices are not open to researchers, investigating the validity is difficult, as it is not possible to know whether the observed differences are a result of the device itself or, more likely, algorithm differences. 

### 4.4. Future Directions

The RADAR-Base tool allowed us to explore the risk factors for depression across diagnoses under the difficult circumstances caused by the national pandemic restrictions, future uncertainty and fear of the contagion, and fear of one’s death or that of their family members/friends. This situation could have aggravated depression in people who already suffered from depression, increased feelings of loneliness, and reduced physical activity, especially in people with moderate or severe depression. Low HR variation and increased resting mean HR appear to be common pathophysiological factors across diagnoses and might be stress indicators. Future studies must explore how people with different depression severity react in stressful situations.

## Figures and Tables

**Figure 1 jcm-11-07163-f001:**
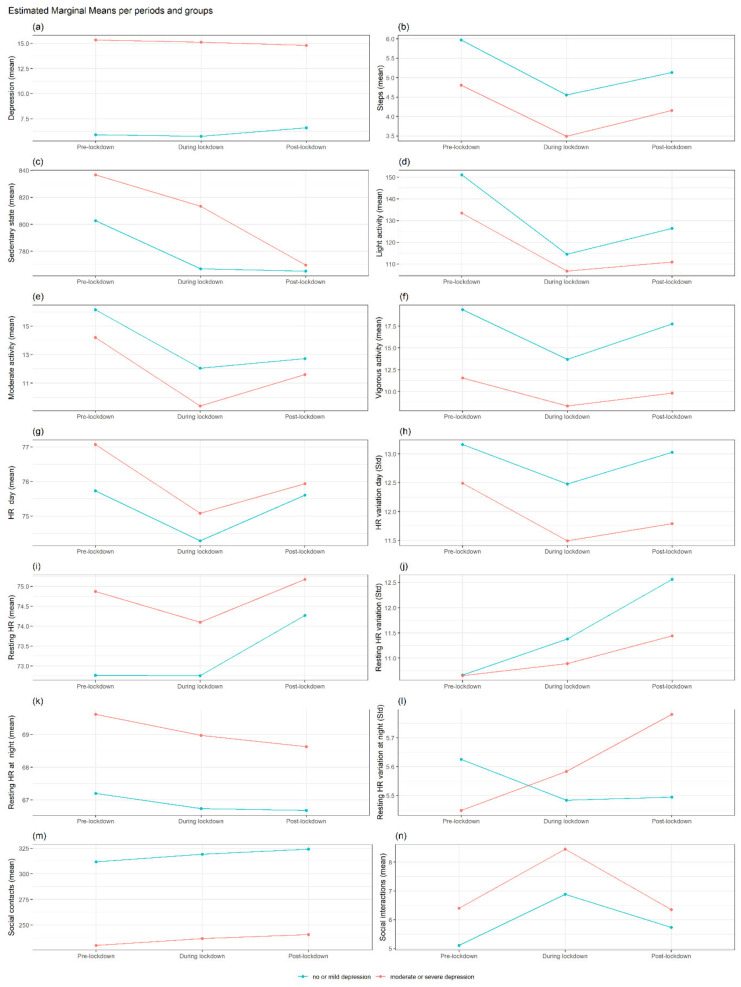
Interaction trajectories by depression status in MDD group. Estimated marginal means per MMD groups based on depression status (moderate or severe depression versus no or mild depression at baseline) across the three periods (pre, lockdown and post-lockdown) for depression severity (**a**), activity features (**b**–**f**), HR features (**g**–**l**), sociality (**m**,**n**). Note: PHQ-8 = depression severity, Steps = mean steps per day, Sedentary = Number of minutes per day classified as sedentary. Light, moderate and vigorous activity = the number of minutes per day classified as “light”, “moderate” or “vigorous activity”, mHR day = mean heart rate (HR) during 24 h, std HR = standard deviation of HR during 24 h, restingHR day = HR at rest during 24 h, Resting HR at night = HR at rest during the night (0:00–05:59), social contacts = number of contacts in the agenda, social interactions = interactions trough social apps.

**Figure 2 jcm-11-07163-f002:**
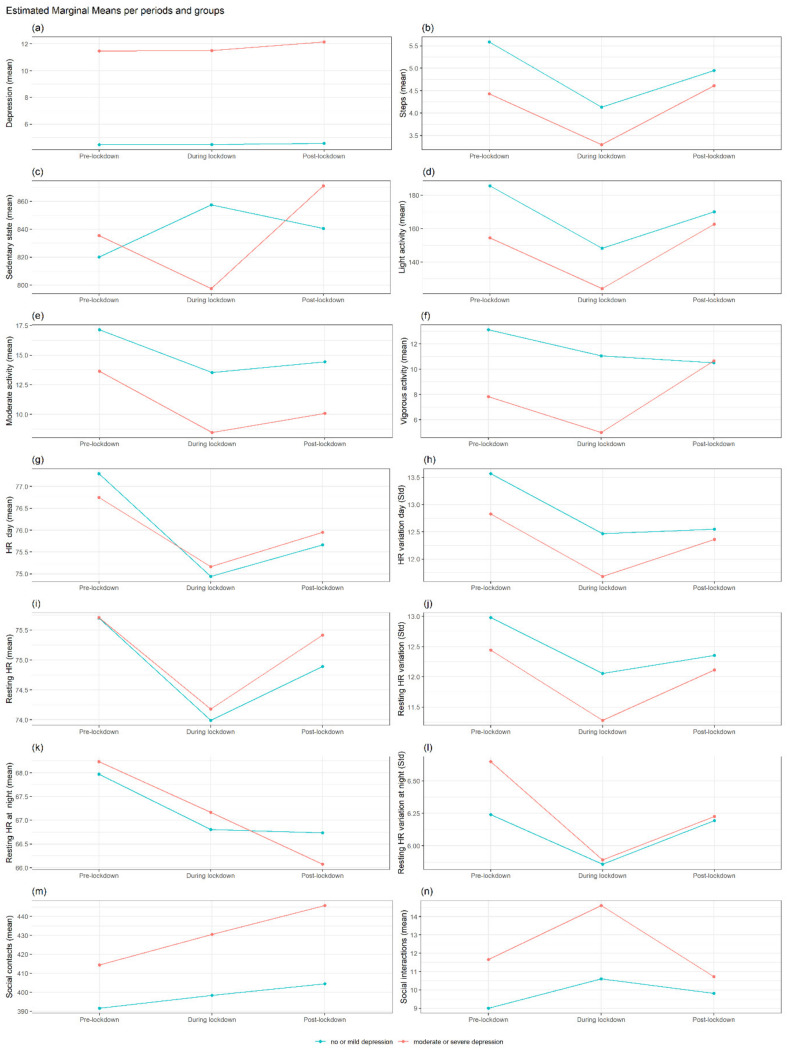
Interaction trajectories by depression status in MS group. Estimated marginal means per MS groups based on depression status (moderate or severe depression versus no or mild depression at baseline) across the three periods (pre, lockdown and post-lockdown) for depression severity (**a**), activity features (**b**–**f**), HR features (**g**–**l**), sociality (**m**,**n**). Note: PHQ-8 = depression severity, Steps = mean steps per day, Sedentary = Number of minutes per day classified as sedentary. Light, moderate and vigorous activity = the number of minutes per day classified as “light”, “moderate” or “vigorous activity”, mHR day = mean heart rate (HR) during 24 h, std HR = standard deviation of HR during 24 h, restingHR day = HR at rest during 24 h, Resting HR at night = HR at rest during the night (0:00–05:59), social contacts = number of contacts in the agenda, **social interactions =** interactions trough social apps.

**Table 1 jcm-11-07163-t001:** Variables assessed in the study.

Clinical and Social Activity Variables Derived from the Apps		
Depression Level	PHQ-8 score	PHQ-8 score last observation before pre-lockdown phase.
		Score range = 0–24
		Cut-off score ≥ 10 moderate to severe depression
Social activity		
Contacts	Daily mean of contacts	Number of all contacts per day
Social interactions	Sum of interactions and short cuts with the social apps	Number of interactions with social apps (whatsapp, facebook, twitter and others) used to make calls, read and send messages in one day
Physiological features (derived from Fitbit device)		
Physical activity		
Heart rate parameters		
Mean HR day	Mean HR during the day	Mean HR across whole day (24 h)
HR variation /day	Std of HR during the day	Standard deviation of HR across whole day (24 h)
Resting mean HR day	Mean HR during the resting period across whole day	Resting of mean HR across whole day (24 hours)
Resting HR variation / day	Std of HR during the resting period across whole day	Std of HR during sedentary periods, identified by activity level = sedentary periods and number of steps = 0 (24 h)
Resting mean HR at night	Mean HR at rest during the night	Mean HR during sedentary periods, identified by activity level = sedentary and number of steps = 0 only during night time (0:00–05:59)
Resting HR variation at night	Std of HR variation at night	Standard deviation of HR
Steps	Mean steps	Mean steps per minute all day (24 h)
Sedentary state	“Sedentary” Minutes	The number of minutes in a day for which the physical activity was classified as “sedentary”.
Light, Moderate and Vigorous activity	“Active” Minutes	The number of minutes in a day for which the physical activity was classified as “light”, “moderate” or “vigorous activity”

**Table 2 jcm-11-07163-t002:** Sample and feature characteristics.

	MDD (N = 255)	MS (N = 214)
Age (Mean, SD)	47.73 (15.42)	44.77 (9.94)
Gender (Women, N %)	190 (74.51)	144 (67.29)
Marital status		
Without partner (Single/ Separated Divorced/Widowed)	129 (50.59%)	73 (34.43%)
(Partner/Married	126 (49.41%)	139 (65.57%)
Education years mean (SD)	15.01 (6.15)	16.91 (5.91)
	** *Median (IQR)* **	** *Median (IQR)* **
PHQ-8	9 (10)	5 (6)
Physical activity		
Steps	6076. (7243)	6523 (6912)
Sedentary	949.36 (385.89)	925.29 (322.57)
Light activity	138.07 (121.89	166.29 (150.07)
Moderate activity	8.36 (17.14)	8.43 (18.14)
Vigorous activity	6.71 (20.86)	3.86 (11.71)
HR parameters		
mean HR day	74.87 (11.22)	76.72 (9.86)
std HR day	12.46 (3.95)	12.87 (4.11)
Resting mean HR day	72.82 (11.03)	75.44 (9.79)
Resting std HR day	11.14 (4.18)	12.42 (3.79)
Resting mean HR at night	66.53 (11.66)	67.04 (10.58)
Resting std HR at night	4.93 (2.15)	5.43 (3.03)
Social activity parameters		
Social contacts	136 (163)	258 (344)
Social interactions	5 (10)	6 (13)

Note: Note: PHQ-8 = depression severity Steps = mean steps per day, Sedentary = The number of minutes per day classified as sedentary. Light, moderate and vigorous activity = the number of minutes per day classified as “light”, “moderate” or “vigorous activity”, mHR day = mean HR during 24 h, std HR = standard deviation of HR during 24 h, restingHR day = HR at rest during 24 h, Resting HR at night = HR at rest during the night (0:00–05:59), social contacts = number of contacts in the agenda, social interactions **=** interactions trough social apps.

**Table 3 jcm-11-07163-t003:** Estimated mean differences in each outcome between periods in MDD (Linear mixed-effects model).

	Pre- and During Lockdown (95% CI, *p*-Value)	Pre- and Post-Lockdown 95% CI, *p*-Value)	During and Post-Lockdown 95% CI, *p*-Value)
**PHQ-8**	0.24 (−0.32 to 0.81)	0.03 (−0.58 to 0.64)	−0.22 (−0.78 to 0.35
**Steps**	**1.32 (0.91 to 1.73) *****	**0.79 (0.33 to 1.26) ****	**−0.52 (−0.99 to −0.061) *****
**Sedentary**	**52.0 (0.59 to 103.4) ***	**73.7 (16.59 to 130.8) ****	21.7(−28.040 to 71.5)
**Light activity**	**38.06 (25.9 to 50.21) *****	**30.35 (17.1 to 43.55) *****	−7.71(−19.4 to 3.99)
**Moderate activity**	**4.24 (1.97 to 6.51) *****	**2.94 (0.32 to 5.56) ***	−1.30 (−3.02 to 0.42)
**Vigorous activity**	**3.45 (1.29 to 5.61) *****	1.31 (−0.93 to 3.55)	**−2.14 (−4.28 to −0.004) ***
**Mean HR day**	**1.38 (0.78 to 1.99) *****	0.43 (−0.29 to 1.14)	**−0.95 (−1.69 to −0.22) ****
**std HR day**	**0.75 (0.37 to 1.14) *****	**0.44 (0.039 to 0.84) ***	−0.32 (−0.77 to 0.13)
**Resting mean HR day**	−0.037 (−0.64 to 0.57)	**−1.25 (−1.98 to −0.51) *****	**−1.21 (−1.94 to −0.48) *****
**Resting std HR day**	**−0.72 (−1.24 to −0.19) ****	**1.45 (−2.00 to 0.93) *****	**−0.75 (−1.23 to 0.27) *****
**Resting mean HR at night**	0.31 (−0.40 to 1.03)	0.53 (−0.23 to 1.29)	0.21 (−0.39 to −0.82)
**Resting std HR at night**	0.065 (−0.21 to 0.34)	−0.06 (−0.38 to 0.26)	−0.12 (−0.44 to 0.19)
**Social contacts**	**−7.30 (−12.6 to −2.04) ******	**−11.85 (−19.31 to −4.36) ****	**−4.55 (−65.8 to −56.72) ***
**Social interactions**	**−2.42 (−3.64 to −1.20) *****	**−1.05 (−2.11 to 0.01) *****	**1.37 (0.138 to 2.60) ***

**Note:** PHQ-8 = depression severity, mHR day = mean HR during 24 h, std HR = standard deviation of HR during 24 h, resting mHR day = mean HR at rest during 24 h, Activity mHR day = mean HR during activity period during 24 h, Resting mHR at night = mean HR at rest during the night (0:00–05:59), Steps = mean steps per day, Sedentary = The number of minutes per day classified as sedentary. Light, moderate and vigorous activity = the number of minutes per day classified as “light”, “moderate” or “vigorous activity”, social contacts = number of contacts in the agenda, social interactions = interactions trough social apps. * *p* < 0.05; ** *p* < 0.01; *** *p* < 0.001.

**Table 4 jcm-11-07163-t004:** Estimated mean differences in each outcome between periods in MS (Linear mixed-effects model).

	Pre- and During Lockdown (95% CI, *p*-Value)	Pre- and Post-Lockdown 95% CI, *p*-Value)	During and Post-Lockdown 95% CI, *p*-Value)
**PHQ-8**	−0.07 (−0.62 to 0.47)	−0.191 (−0.79 to 0.41)	−0.117 (−0.66 to 0.43)
**Steps**	**1.36 (0.82 to 1.89) *****	0.44 (−0.04 to 0.93)	**−0.91 (−1.35 to −0.48) *****
**Sedentary**	−7.33 (−74.3 to 59.7)	−4.21 (−76.0 to 67.6)	3.12 (−63.4 to 69.6)
**Light activity**	**35.5 (17.0 to 53.9) *****	**12.7 (−9.0 to 34.49)**	**−22.7 (−41.8 to −3.67) ***
**Moderate activity**	**3.72 (0.069 to 7.36) ***	2.71 (−0.90 to 6.33)	−1.01 (−4.30 to 2.30)
**Vigorous activity**	2.17 (−1.11 to 5.46)	1.14 (−2.22 to 4.49)	−1.04 (−5.01 to 2.93)
**mHR day**	**2.14 (1.26 to 3.02) ****	**1.38 (0.44 to 2.32) ****	−0.76 (−1.57 to 0.05)
**std HR day**	**1.10 (0.56 to 1.63) *****	**0.80 (0.32 to 1.28) ****	−0.30 (−0.85 to 0.25)
**Resting mHR day**	**1.67 (0.78 to 2,57) *****	0.64 (−0.28 to 1.57)	**−1.03 (−1.85 to −0.21) ****
**Resting std HR day**	**0.98 (0.48 to 1.49) *****	**0.46 (0.014 to 0.91) ***	−0.53 (−1.07 to 0.020
**Resting mHR at night**	**1.20 (0.02 to 2.39) ***	**1.37 (0.16 to 2.58) ****	0.16 (−1.04 to 1.37)
**Resting std HR at night**	**0.57 (0.02 to 1.15) ***	0.12 (−0.29 to 0.53)	−0.44 (−1.01 to 0.11)
**Social contacts**	**−9.17 (−15.7 to −2.61) ****	−18.01 (−44.4 to 8.40)	−8.84 (−37.2 to 19.55)
**Social interactions**	−1.92 (−4.15 to 0.31)	−1.01 (−3.46 to 1.44)	0.91 (−2.33 to 4.14)

**Note:** PHQ-8 = depression severity, mHR day = mean HR during 24 h, std HR = standard deviation of HR during 24 h, resting mHR day = mean HR at rest during 24 h, Activity mHR day = mean HR during activity period during 24 h, Resting mHR at night = mean HR at rest during the night (0:00–05:59), Steps = mean steps per day, Sedentary = The number of minutes per day classified as sedentary. Light, moderate and vigorous activity = the number of minutes per day classified as “light”, “moderate” or “vigorous activity”, social contacts = number of contacts in the agenda, social interactions = interactions through social apps. * *p* < 0.05; ** *p* < 0.01; *** *p* < 0.001.

## Data Availability

The datasets used and/or analyzed during the current study are available from the corresponding author on reasonable request.

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
