# Peer review of "Biopsychosocial Response to the COVID-19 Lockdown in People with Major Depressive Disorder and Multiple Sclerosis"

_jcm, 2022, doi:10.3390/jcm11237163_

Round 1

Reviewer 1 Report

1.       What are the future implications of this study? Or how the study will contribute to the science? Please highlight it in the abstract.

2.       What was the time duration of the study?

3.       Did the authors consider the effects of vaccination?

4.       What was the medication status of patients during the study (for covid19)?

5.       What was the medication status of patients during the study (for MS and MDD)?

6.       Figure legends should be detailed.

7.       What about the statistical significance in the figures?

Author Response

  1. What are the future implications of this study? Or how the study will contribute to the science? Please highlight it in the abstract.

Reply: Conclusions: Behavioral response to the lockdown measured by social activity, physical activity and HR may reflect changes in stress in people with MDD and MS.  Future studies must explore how people with different depression severity react in stressful situations. Remote technology monitoring might promptly activate an early warning of physical and social alterations in these situations.

  1. What was the time duration of the study?

Reply: from December, 2019 until June 2020

  1. Did the authors consider the effects of vaccination?

Reply: the effect of vaccination has not started yet during the period included in the study. Moreover, it was not relevant to the study objective. The main goal was to explore the impact of the Covid-19 lockdown, not the Covid-19 infection.

  1. What was the medication status of patients during the study (for covid19)?

Reply: Patients were under antidepressant treatments. The main objective was to explore the impact of the Covid-19 lockdown, not the Covid-19 infection.

  1. What was the medication status of patients during the study (for MS and MDD)?

Reply: Patients were under standard treatments.

  1. Figure legends should be detailed.

Reply: added figure legends

Figure 1. Interactions trajectories by depression status in MDD group

Estimate marginal means per groups of MDD with different depression status (moderate or severe depression versus N=124 no or mild depression at baseline N=129) across the three periods (pre, lockdown and post-lockdown) for each depression level (a), activity features (b-f), HR features (g-l), sociality (m,n).  

Note: PHQ-8= depression severity Steps= mean of steps per day, Sedentary = The number of minutes per day classified as a sedentary. Light, moderate and vigourous activity: the number of minutes per day classified as “lightly” or “moderate” and “vigorous “activity”, , mHR day=mean HR during 24h, std HR = standard deviation of HR during 24h, restingHR day= HR at rest during 24h, Resting HR at night= HR at rest during the night (0:00-05:59), social contacts = number of contacts in the agenda, social interactions = interaction trough the social apps.

Figure 2. Interactions trajectories by depression status in MS group

Estimate marginal means per groups of MS with different depression status (moderate or severe depression versus N=53 no or mild depression at baseline N=152) across the three periods (pre, lockdown and post-lockdown) for each depression level (a), activity features (b-f), HR features (g-l), sociality (m,n).

Note: PHQ-8= depression severity Steps= mean of steps per day, Sedentary = The number of minutes per day classified as a sedentary. Light, moderate and vigourous activity: the number of minutes per day classified as “lightly” or “moderate” and “vigorous “activity”, , mHR day=mean HR during 24h, std HR = standard deviation of HR during 24h, restingHR day= HR at rest during 24h, Resting HR at night= HR at rest during the night (0:00-05:59), social contacts = number of contacts in the agenda, social interactions = interaction trough the social apps.

  1. What about the statistical significance in the figures?

Reply:  Mean depression trajectories by depression status and gender between pre-lockdown and during lockdown, interaction term p=0.027

Resting mean HR at night trajectories by depression status and gender between pre-lockdown and post-lockdown, interaction term p= 0.047

Reviewer 2 Report

I have reviewed "Biopsychosocial response to the COVID-19 lockdown in people with major depressive disorder and multiple sclerosis" by Sara Siddi et al., a multi-center study funded from the EU's Horizon funds that aimed to explore how the lockdown impacts two different health conditions by depression severity. The manuscript is written in proper English, the abstract and introduction are sufficient. Methodology - how did you pick the country to analyze MS and MDD patients from? Why MDD patients come from NL, ES and UK and MS from IT, DM, ES? Please elaborate to make this more clear. I see you mentioned that different economic status and other factors could affect the results in the limitations so that is sufficient but the extra information in methodology could be useful. (141-143) Results are clearly presented. Discussion is sufficient, good limitations section is a strength of this manuscript. I have no ethical comments since CoI is very clear. I also have no other comments, this is a valid study and the authors obviously knew how to prepare a manuscript. Congratulations, an interesting read.

Author Response

  1. I have reviewed "Biopsychosocial response to the COVID-19 lockdown in people with major depressive disorder and multiple sclerosis" by Sara Siddi et al., a multi-center study funded by the EU's Horizon funds that aimed to explore how the lockdown impacts two different health conditions by depression severity. The manuscript is written in proper English, the abstract and introduction are sufficient.

Reply: thank you for the commendation.

  1. Methodology - how did you pick the country to analyze MS and MDD patients from? Why MDD patients come from NL, ES and UK and MS from IT, DM, ES? Please elaborate to make this more clear.

Reply: This study is part of the RADAR-CNS consortium that included three mental health conditions. The information was reported in “ 2. Materials and Methods

Digital data were collected as part of the international research consortium RADAR-CNS (https://www.radar-cns.org/), a decentralized collaborative research initiative aiming to provide real-time multidimensional indicators of the clinical states of individuals with different health conditions: MDD, MS and epilepsy. This manuscript, we focused our analyses on two health conditions: MDD [41] and MS [42]. Participants were recruited in 5 European countries: people with Major Depressive Disorder (Netherlands, Spain and the UK) and people with Multiple Sclerosis (Italy, Denmark and Spain). The recruitment and follow-up started in November 2017 and finished in March.”

  1. I see you mentioned that different economic status and other factors could affect the results in the limitations so that is sufficient but the extra information in methodology could be useful. (141-143)

Reply:

We also studied factors that might influence the behavioral and social features during the lockdown period. The investigated factors included age, gender, marital status and education in years.

Results are clearly presented. Discussion is sufficient, good limitations section is a strength of this manuscript. I have no ethical comments since CoI is very clear. I also have no other comments, this is a valid study and the authors obviously knew how to prepare a manuscript. Congratulations, an interesting read.

Reply: thank you again for you effort in revising the manuscript.

Reviewer 3 Report

This paper states an interesting research question - biopsychological effects of COVID-19 lockdown as experienced in multiple countries, and aims to answer it using both active and passive remote assessment data. I find that the aim and selected method has a lot of potential, but unfortunately the paper does not fulfill this promise.

I do not understand the choice of MDD and MS patient groups, other than being convenience samples, which authors state on lines 97-98 is a problem in previous COVID-19 research.

Further, while analyses take into account assessment of depression symptoms pre-lockdown, there is no controlling for treatment (pharmaceutical or therapy) or changes in treatment. It seems to me that an MDD population would be managed quite well pre-pandemic and continue treatment during lockdown (perhaps switched to virtual sessions rather than in-person, if therapy) and that not much variation in depressive symptoms should be observed in this population. 

The lockdown circumstances in different countries (data collection sites) need to be presented much more in detail and problematized. (lines 158-165)

Overall, language is confusing and unclear. Examples:
Table 1. What is meant by std of HR? How is "across all day" defined? 24 hours? Or is night-time excluded? 
Table 2. mean steps per day is 4.22 and 4.53, which is impossible. Do you mean 4.22*10^3 ?
Line 106 "vast quantity of data over period"
Line 252 "we included a three-way interaction period for baseline depression severity and gender, as exposure variables, to investigate whether the rate of change in the outcome variables over time varied according to depression severity and gender" This is not a 3-way interaction, it is a 2-way interaction. Further, the presented results in Table 3-4 and Figures 1-2 do not measure 'rate of change in outcome variables', which signals to me that the authors do not fully understand the methods used and/or the correct choice of method to answer such research questions.

Unfortunately, the paper overall gives a very hurried impression.

Author Response

  1. This paper states an interesting research question - biopsychological effects of COVID-19 lockdown as experienced in multiple countries, and aims to answer it using both active and passive remote assessment data. I find that the aim and selected method has a lot of potential, but unfortunately the paper does not fulfill this promise. I do not understand the choice of MDD and MS patient groups, other than being convenience samples, which authors state on lines 97-98 is a problem in previous COVID-19 research.

Reply:  This study is part of the RADAR-CNS international research project where the main objective is to develop ways of measuring major depressive disorder, epilepsy multiple sclerosis using wearable devices and smartphone technology during a follow-up of 2 years. 

The data collection started from November 2017 until April 2021, so included the COVID-19 lockdown period.  In this paper, by leveraging the data already collected previously, COVID-19 offers us the opportunity to study the biopsychosocial impact of the COVID-19 lockdown in our sample composed of MDD and MS using an innovative platform RADAR-Base that collects a massive quantity of data.  

Contrary to other studies that included convenient participants for COVID-19 projects without having a measure of their biopsychosocial state before COVID-19.  Furthermore, we conducted the analysis to see the differences in depression severity and gender.

  1. Further, while analyses take into account assessment of depression symptoms pre-lockdown, there is no controlling for treatment (pharmaceutical or therapy) or changes in treatment. It seems to me that an MDD population would be managed quite well pre-pandemic and continue treatment during lockdown (perhaps switched to virtual sessions rather than in-person, if therapy) and that not much variation in depressive symptoms should be observed in this population.

Reply:   Participants were under standard medication during the study. 

We have reported as follows in the introduction “Other studies found an increase in depression severity in patients with MDD caused by the pandemic, supported by higher hospitalizations of patients with MDD at a short-term acute care hospital and all-cause inpatient hospitalizations at a psychiatric facility increased following the pandemic onset. and remained elevated throughout the first six months” 

The virtual sessions probably increased during the lockdown and replaced in-person therapy. Still, the satisfaction of using them might depend on the higher levels of e-health literacy and being comfortable in sharing their feelings and thoughts in the family's presence during the lockdown.

Reference:  Liberman JN, Bhattacharjee S, Rui P, Ruetsch C, Rothman B, Kulkarni A, Forma F. Impact of the COVID-19 Pandemic on Healthcare Resource Utilization in Individuals with Major Depressive Disorder. Health Serv Res Manag Epidemiol. 2022 Jul 6;9:23333928221111864. doi: 10.1177/23333928221111864. PMID: 35832488; PMCID: PMC9272161.

  1. The lockdown circumstances in different countries (data collection sites) need to be presented much more in detail and problematized. (lines 158-165)

Reply:  We have detailed this part: 

The regional and local impact of the COVID-19 crisis has been highly heterogeneous, with significant implications for crisis management and policy responses [44] causing physical distance, loss of familiar and social connections in person and worsening mental health.[45].Different restrictions were implemented across countries with the requirement of staying at home except for essential trips for work and all public events were cancelled, except Denmark, where the measure was only recommended. Italy and Spain had the more restrictive measures compared to the other countries [46] restrictions on gatherings and school closures (only geographically targeted). Places of work were required to close in some sectors in Spain, the United Kingdom, and Denmark, and were required to close for all but public services (stores, libraries, museum etc.) in the Netherlands and Italy. Public transport was recommended to close in Italy, Spain, and Denmark [14].

However the differences by countries of these data were previously explored

Sun S, Folarin AA, Ranjan Y, Rashid Z, Conde P, Stewart C, Cummins N, Matcham F, Dalla Costa G, Simblett S, Leocani L, Lamers F, Sørensen PS, Buron M, Zabalza A, Guerrero Pérez AI, Penninx BW, Siddi S, Haro JM, Myin-Germeys I, Rintala A, Wykes T, Narayan VA, Comi G, Hotopf M, Dobson RJ; RADAR-CNS Consortium. Using Smartphones and Wearable Devices to Monitor Behavioral Changes During COVID-19. J Med Internet Res. 2020 Sep 25;22(9):e19992. doi: 10.2196/19992. PMID: 32877352; PMCID: PMC7527031.

  1. Overall, language is confusing and unclear. Examples:

Table 1. What is meant by std of HR? How is "across all day" defined? 24 hours? Or is night-time excluded?

Reply: we made the corrections properly. All day means 24 hours.

  1. 5. Table 2. mean steps per day is 4.22 and 4.53, which is impossible. Do you mean 4.22*10^3 ?4

Reply:  Mean of steps per minute during 24 hours. We converted in minutes per day,  Mean*60 minutes* for 24 hours.

  1. Line 106 "vast quantity of data over period"

Reply: thank you, we have changed for large, we have also corrected English mistakes.

  1. 7. Line 252 "we included a three-way interaction period for baseline depression severity and gender, as exposure variables, to investigate whether the rate of change in the outcome variables over time varied according to depression severity and gender" This is not a 3-way interaction; it is a 2-way interaction. Further, the presented results in Table 3-4 and Figures 1-2 do not measure 'rate of change in outcome variables', which signals to me that the authors do not fully understand the methods used and/or the correct choice of method to answer such research questions.

Reply:  We regret that we might have not explained ourselves properly.  Tables 3 and 4 represent the model and show the findings of the linear mixed-effects regression analyses for repeated measures (depressive symptoms, HR, physical activity and social activity) to assess the changes in the mean total score of each daily parameter among the three periods: pre-, during and post-lockdown (defined for each country), without interaction term.  We included in the following models baseline depression severity and gender, as exposure variables, including an interaction term between period, depression severity and gender, to investigate whether the rate of change in the outcome variables over time varied according to depression severity and gender

These findings are represented in tables s2 and s3 in the supplementary materials.  While Figures 1 and 2 show the findings according to the depression severity and period.

Round 2

Reviewer 1 Report

The authors successfully responded to the reviewer's comments and updated the manuscript as well. 

Reviewer 3 Report

The authors have made efforts to address the concerns I stated. Nonetheless, the paper would still benefit greatly from English language improvements.